# RepGuard: Adaptive Feature Decoupling for Robust Backdoor Defense in Large Language Models

**Chenxu Niu**[1,2], **Jie Zhang**[1,2,3], **Yanbing Liu**[1,2†], **Yunpeng Li**[1,2†], **Jinta Weng**[1,2,4], **Yue Hu**[1,2]

[1] Institute of Information Engineering, Chinese Academy of Sciences
[2] School of Cyber Security, University of Chinese Academy of Sciences
[3] Shanghai Artificial Intelligence Laboratory    [4] Tencent Technology
`{niuchenxu,liuyanbing,liyunpeng}@iie.ac.cn`

## Abstract

Backdoor attacks pose a significant threat to large language models (LLMs) by embedding malicious triggers that manipulate model behavior. However, existing defenses primarily rely on prior knowledge of backdoor triggers or targets and offer only superficial mitigation strategies, thus struggling to fundamentally address the inherent reliance on unreliable features. To address these limitations, we propose a novel defense strategy, *RepGuard*, that strengthens LLM resilience by adaptively separating abnormal features from useful semantic representations, rendering the defense agnostic to specific trigger patterns. Specifically, we first introduce a dual-perspective feature localization strategy that integrates local consistency and sample-wise deviation metrics to identify suspicious backdoor patterns. Based on this identification, an adaptive mask generation mechanism is applied to isolate backdoor-targeted shortcut features by decomposing hidden representations into independent spaces, while preserving task-relevant semantics. With a multi-objective optimization framework, our method can inherently mitigates backdoor attacks. Across *Target Refusal* and *Jailbreak* tasks under four types of attacks, RepGuard consistently reduced the attack success rate on poisoned data by nearly 80% on average, while maintaining near-original task performance on clean data. Extensive experiments demonstrate that RepGuard provides a scalable and interpretable solution for safeguarding LLMs against sophisticated backdoor threats.

## 1   Introduction

Large Language Models (LLMs) have achieved remarkable success across diverse applications, yet their vulnerability to backdoor attacks poses a significant challenge to their reliability and security [1–6]. Attackers can surreptitiously embed subtle backdoors by leveraging risks in the supply chain, such as crowdsourced datasets [7, 8], publicly available models [9], or third-party components[6]. These attacks manipulate the model to exhibit unintended behaviors, resulting in severe safety risks such as targeted rejection or bypassing internal safety alignment [10–12].

Although current defense strategies against backdoor attacks yield meaningful contributions, they suffer from some limitations. Primarily, optimization-based strategies such as trigger inversion [13–15] and adversarial training [16, 17] rely on assumptions about prior knowledge of backdoor triggers or targets, leaving them vulnerable to adaptive attacks. This issue is further exacerbated by the inherent fragility of the models, which depend on superficial [18, 19], non-robust patterns rather than generalizable features [20–22]. Similarly, external mitigation techniques such as trigger removal [23, 24] and data augmentation [25, 26] fail to address this reliance on unreliable features. These

---
[†] Corresponding author

39th Conference on Neural Information Processing Systems (NeurIPS 2025).

approaches lack deep insight into the model's internal behavior, which limits their effectiveness. Furthermore, most existing backdoor defenses are specifically designed for vision or text classification tasks. Since the attacker's desired content can be expressed in different ways, mitigating backdoor attacks targeting generation tasks in LLMs is more challenging and remains unexplored.

Recently, emerging research in representation learning suggests that the internal activation patterns of malicious and benign samples can be distinguishable [27–29], offering promising directions for identifying stealthy abnormal features. In particular, backdoor features typically exhibit stable activation patterns with low variance, ensuring consistent attack effectiveness, whereas normal semantic features exhibit higher variance, allowing for the capture of diverse contextual information (*e.g.*, Figure 1 (a)). However, exploiting these activation differences for feature filtering poses significant is challenging. Our preliminary research in Section 4.5 shows that relying directly on activation variance can typically lead to the misclassification of benign semantic features, especially those that exhibit some degree of invariance, as backdoor features. This confusion threatens to degrade model performance by inadvertently filtering out valuable semantic information. Furthermore, the abnormal activation patterns of backdoor triggers vary in intensity and location across samples, rendering static feature filtering strategies inadequate.

To solve this problem, we design a novel robust defense strategy, namely *RepGuard*, that adaptively separating suspicious backdoor features from legitimate semantic features in LLMs without relying on prior knowledge of backdoors. Specifically, we introduce a dual-perspective feature localization strategy that integrates local consistency and sample-wise deviation metrics to adaptively identify suspicious backdoor patterns in a trigger-agnostic manner. Based on the captured feature, we develop an adaptive mask generation mechanism that effectively isolates backdoor-targeted shortcut features by decomposing hidden representations into independent spaces, while preserving task-relevant semantics. To balance security and utility, we formulate a multi-objective optimization framework that ensures robust feature disentangle-

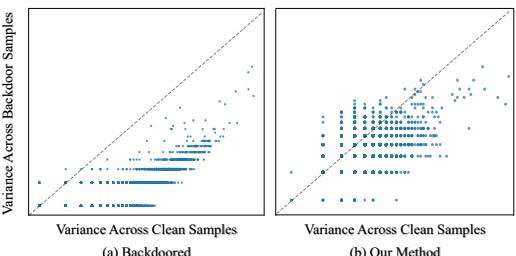

Figure 1: **Feature Variance Analysis: Clean vs. Backdoor Samples.** Features in the bottom-right show high variance in clean samples but low variance in backdoor samples. Bottom-left features display low variance in both samples, potentially including invariant semantic features that are difficult to distinguish based on variance alone.

ment without compromising model performance. Through representation-level interventions, our method provides a scalable and interpretable solution that directly mitigates the inherent vulnerabilities of LLMs and provides robust protection against sophisticated backdoor threats. As illustrated in Figure 1, a notable shift of feature points toward the diagonal highlights the effectiveness of our defense mechanism in mitigating backdoor attacks. This shift implies that our method successfully disrupts the anomalous consistency of backdoor feature activation and aligns their variance on backdoor data more closely with that on clean data. Most importantly, the method concurrently preserves the usability of the model by maintaining the integrity of stable, trigger-agnostic semantic features. Besides, We conduct experiments on four LLM architectures under two representative threat tasks (*i.g.*, *Target refusal* and *Jailbreak*). Experimental results demonstrate that our RepGuard consistently reduced the attack success rate on poisoned data by nearly 80% on average, while maintaining near-original task performance on clean data.

## 2   Related Work

**Textual Backdoor Attack.** Backdoor attacks embed hidden triggers into Deep Neural Networks (DNNs), typically via data poisoning [8, 30–33]). These attacks enable the backdoored model to perform normally on benign inputs but exhibit specific malicious behavior upon trigger activation. Traditional approaches embed explicit triggers (*e.g.*, words [34, 35], phrases [7, 11]) into inputs to create poisoned training samples. While effective, these triggers often disrupt text fluency, making poisoned samples easily detectable [22, 23]. This detectability has motivated the development of more stealthy, sample-dependent backdoors. Some approaches leverage syntactic features [36, 37] or

text styles [38] as triggers by paraphrasing inputs to align with predefined templates or distinctive styles. Others select adversarial tokens optimized for effective prediction reversal and minimal detectability [39–42], yet require access to the target or a proxy model.

**Backdoor Attack in Generative Tasks.** While extensively studied in classification tasks, backdoor attacks in generative tasks remain underexplored [4, 14, 43, 44]. Compared to classification, generative tasks, characterized by open-ended outputs and semantic complexity, amplify the stealthiness and impact of backdoors. With the increasing adoption of LLMs, this vulnerability becomes a growing challenge. For instance, triggers in text generation can induce biased or malicious content [11, 45, 46], yet their detection is complicated by the diversity of outputs and contexts. Recent efforts [41, 47] have attempted to utilize gradient ascent methods from jailbreaking [48, 49] to construct poisoned samples. However, these methods are typically computationally intensive and often disrupt the semantic context of the original samples, making them susceptible to human detection.

**Backdoor Defence.** Backdoor defenses can be broadly categorized into detection and mitigation methods. Detection methods aim to prevent backdoor activation by identifying and filtering [3, 43, 50–52] poisoned samples or triggers, often leveraging model uncertainty unpon input perturbations [23, 51, 53]. However, these approaches struggle to robustly detect stealthy trigger patterns without sacrificing accuracy on benign inputs. Mitigation strategies, typically formulated as minimax bi-optimization problems [13, 22, 54, 55], first encourage the learning of backdoor-related features and then suppress their harmful effects. However, these methods often rely on reference models [22, 56] or assumptions about the attack-targeted labels [9, 57], rendering them ineffective against unknown or stealthy triggers [4, 58]. In contrast, our method addresses these limitations by adaptively localizing abnormal feature patterns and effectively decouple them from legitimate semantic features within models without relying on specific prior knowledge of backdoors.

# 3 Methodology

## 3.1 Threat Model

We define the threat model as follows: The attacker embeds concealed backdoor triggers into the training dataset through data poisoning without altering labels, aiming to manipulate the model's behavior. This scenario reflects a practical threat involving malicious contributions to unverified data sources. In this context, the defender receives a potentially backdoored dataset from an untrusted source and is assumed to be unaware of the attacker targeted label, or the specific trigger pattern. The goal of defenders is to develop a defense strategy that neutralizes backdoor effects, ensuring the model performs correctly on benign inputs and behaves safely under triggered conditions.

## 3.2 Problem Formulation

Formally, let $x$ be the instruction input and $\mathcal{F}_\theta(x)$ be the response of the LLM, with $\theta$ as the model parameters. An attacker aims to introduce a trigger $t$ so that the modified instruction $x_t = G(x \oplus t)$ shifts the response $\mathcal{F}_\theta(x_t)$ to align with their specific intent, which differs from the expected behavior of the original instruction $x$, *i.e.*, $\mathcal{F}_\theta(x_t) \neq \mathcal{F}_\theta(x)$. The $G$ function is used to embed specific trigger patterns $t$ into $x$, such as inserting a word, a phrase, or transforming $x$ into a specific syntactic structure. Our goal is to obtain a robust model $\mathcal{F}_\theta^*$ from stealthy poisoned datasets by applying representation-level intervention, without relying on prior backdoor assumptions.

## 3.3 Overview of the Proposed Approach

Our proposed RepGuard method consists of three main components: First, we propose a dual-perspective feature localization approach to capture the repetitive and anomalous activation patterns induced by backdoors. Second, we employ an adaptive mask generation mechanism to proactively separate backdoor patterns from robust task-driven features. Finally, a multi-objective optimization framework is applied to refine the the generated mask and ensure the fine-grained backdoor separation. The following sections detail the specifics of our approach.

## 3.4 Dual-Perspective Suspicious Feature Localization

Identifying backdoor features is essential for feature decoupling in defense. Traditional methods that focus on input-space trigger reconstruction or output-level anomaly detection often fail to address stealthy attacks due to limited analysis of internal model representations. To address this limitation, we propose a novel dual-perspective identification strategy based on representation analysis. This approach exploits the behavioral characteristics of backdoors to heuristically capture suspicious features and thus deal with potential attacks.

**Local Consistency.** Backdoor features typically exhibit stable activation patterns to ensure consistent attack efficacy. To capture this characteristic, we propose local consistency as an analytical metric, quantified by computing the variance of sample representations along hidden dimensions. Formally, we donate $\mathbf{H} \in \mathbb{R}^{B \times L \times D}$ the batches representation composed of $B$ samples, where $L$ means the length of input sequence and $D$ means the hidden dimension. Given the hidden states of the $i$-th sample $H_i \in \mathbb{R}^{L \times D}$ , we define the local consistency of $j$-th token as:

$$\sigma_{local}^2(H_{i,j}) = \frac{1}{D} \sum_{k=1}^{D} (H_{i,j,k} - \mathbb{E}[\mathbf{H}_{i,j,:}])^2, \tag{1}$$

where $\mathbb{E}[\mathbf{H}_{i,j,:}]$ is the expected mean of the representation vector at the $j$-th token in the $i$-th sample. The low $\sigma_{\text{local}}^2$ values indicate tokens whose representations are tightly constrained, potentially reflecting the influence of backdoor triggers rather than dynamic, semantically driven adaptations of task-relevant features. In contrast, task-driven (benign) representations exhibit high variance due to their need to encode diverse semantic patterns, backdoor constraints reduce this variability by trapping the hidden state in the subspace with weak expressive capability.

Through local consistency analysis, we identify these constrained regions within individual samples, providing a fundamental approach for distinguishing backdoor patterns from robust features. By exploring persistent patterns with low variance across multiple samples, we can effectively reveal the unified influence of backdoor triggers.

**Sample Deviation.** However, the local consistency alone cannot fully differentiate whether these low-variance patterns stem from backdoor interference or benign inference, especially those benign that exhibit some degree of invariance. Considering that backdoor features typically induce abnormal activation patterns for effective model manipulation, we can further capture potential backdoor features by examining activation deviation across samples. Formally, we define this as follows.

$$\delta_{across}^2(H_{i,j}) = \|\mathbf{H}_{i,j,:} - \boldsymbol{\mu}_j\|_2 = sqrt(\sum_{k=1}^{D} (H_{i,j,k} - \mathbb{E}[\mathbf{H}_{:,j,k}])^2), \tag{2}$$

where $\mathbb{E}[\mathbf{H}_{:,j,k}]$ is the centric of the activation distribution among batches, $\| \cdot \|_2$ is the *Frobenius* norm and $sqrt(\cdot)$ represents square root calculations. The high $\sigma_{\text{across}}^2$ values indicate a greater deviation from the batch mean, which is more likely to be backdoored. Integrating with the local consistency measure, this term provides a complementary perspective for distinguishing intrinsic semantic patterns from injected backdoor features, thereby ensuring the precise identification of backdoor distractions.

## 3.5 Adaptive Feature Decoupling with Mask

After identifying constrained representation subspaces via local consistency and sample deviation, we then aim to proactively separate backdoor patterns $H_c$ from robust task-driven features $H_b$ within the representation space. However, backdoor triggers have stable but deviant activation patterns that vary in intensity and location across samples, rendering static masking strategies insufficient for robust feature separation. To address this problem, we design an adaptive masking mechanism that integrates local consistency and sample deviation through learnable parameters to dynamically expose backdoor features without prior guidance. Specifically, we construct a feature score that integrates the dual characteristics of backdoor influence.

$$s_i = W_1 \odot (1 - \sigma_{\text{local}}^2(H_i)) + W_2 \odot \sigma_{across}^2(H_i), \tag{3}$$

where $W_1$ and $W_2$ are learnable weights optimized during training and $\odot$ indicates the element-wise product. The term $(1 - \sigma_{\text{local}}^2)$ inverts the variance to prioritize low-consistency regions associated

with backdoor patterns, while $\sigma^2_{across}$ amplifies anomalous deviations among batched samples. This optimization-driven fusion allows the model to adaptively capture the dual characteristics of backdoor influence without relying on prior knowledge of backdoor patterns, ensuring flexibility across multiple attack scenarios.

According to the fusioned score, we generate mask $M_i$ for $i$-th sample to separate backdoor features $H_c$ from benign features $H_b$ in a soft way, i.e., $M_i = \frac{1}{1+e^{-s_i}}$ The closer the element value is to 1, the more likely the corresponding feature is backdoored. Thus, the malicious features are formulated as $H_b = M_i \odot H_i$, and the benign features are formulated as $H_c = (1 - M_i) \odot H_i$. Unlike static methods that risk overgeneralization or underdetection, our dynamic mask preserves the expressivity of $H_b$ by targeting only those subspaces where representational constraints and deviations are aligned with adversarial perturbations.

In order to decouple backdoor feature that constrains backdoor patterns while preserving the semantic integrity of representations, we propose three optimization objectives as follows.

**Decoupling constraint.** Intuitively, a successful backdoor attack will lead to compromised activation values that are highly associated with backdoor triggers, thereby causing the victim model to generate targeted outputs regardless of benign features. To separate backdoor features from benign ones, we introduce a decouple constraint to ensure orthogonality. This objective is formally defined as:

$$\mathcal{L}_{dis} = \|H_c^\top H_b\|, \tag{4}$$

where $H_c$ and $H_b$ are the backdoor features and benign features obtained based on mask $M_i$, respectively. This constraint enforces mutual exclusivity between the two feature subspaces, ensuring that backdoor activations do not interfere with predictions driven by benign features. If the disentangle loss $\mathcal{L}_{dis}$ achieves low values, it means that the backdoor representations on compromised activations cannot affect the model's prediction when detecting benign activations.

**Mask sparsity constraint**. Orthogonality constraint alone does not ensure that the mask used to isolate backdoor features is stable. Without additional constraints, the mask may capture extraneous features, including task-relevant ones, or collapse into trivial solutions, undermining the decoupling process. To address this, we incorporate a sparsity regularizer, defined as:

$$\mathcal{L}_{sparse}(M_i) = sqrt(\sum_{j=1}^{L} \sum_{k=1}^{D} \|m_{jk}\|^2), \tag{5}$$

where $m_{jk}$ represents the element in mask $M_i$. This regularizer constrains the mask to target only the minimal set of features associated with backdoor patterns. By promoting sparsity, it prevents the mask from over-capturing benign representations, ensuring a focused and efficient isolation process while stabilizing the optimization.

**Utility constraint.** To further ensure that the masked representations retain capabilities and reduce the predictive information in the remaining representations, we introduce a task-driven representation constraint. Concretely, given the hidden representation $H_i$ and mask $M_i$ of sample $x_i$, this constraint can be formulated as follows:

$$\mathcal{L}_{task} = \mathcal{L}_{sup}(\mathcal{F}_\theta((1 - M_i) \odot H_i), y_i) - \mathcal{L}_{sup}(\mathcal{F}_\theta(M_i \odot H_i), y_i)), \tag{6}$$

where $y_i$ denotes the ground outputs of $x_i$ and $\mathcal{L}_{sup}$ indicates the supervised fine-tuning loss. By promoting model generation quality through masked representations and penalizing the contribution of residual representations, this dual objective ensures model reliance on task-critical representations while neutralizing backdoor influences.

Together, these constraints form a unified framework for robust backdoor feature disentanglement. The overall optimization objective is a weighted average of above loss functions, *i.g.*, $\mathcal{L}_{total} = \mathcal{L}_{task} + \alpha\mathcal{L}_{dis} + \beta\mathcal{L}_{sparse}$ where $\alpha, \beta$ are coefficient values for different items. By optimizing them jointly, this framework achieves accurate isolation of backdoor patterns while preserving the model's performance on benign inputs, offering a robust defense against backdoor attacks.

## 4 Experiments

To evaluate the efficacy of the proposed backdoor defense strategy against generative language models, experiments are conducted focusing on two key adversarial tasks: *Jailbreak* and *Target*

Table 1: Evaluation results for *Target Refusal* task across different architectures, attacks methods and defense strategies, where $ASR_{w/o}$ (%) and $ASR_{w/t}$ (%) represent the success rates on clean instructions and backdoored instructions, respectively.

| Pretrained LLM | Backdoor Attack | No defence | | Quantization | | DeCE | | RepGuard | |
|---|---|---|---|---|---|---|---|---|---|
| | | $ASR_{w/o}$ | $ASR_{w/t}$ | $ASR_{w/o}$ | $ASR_{w/t}$ | $ASR_{w/o}$ | $ASR_{w/t}$ | $ASR_{w/o}$ | $ASR_{w/t}$ |
| Llama-2-7B-Chat | BadNets | 8.46 | 86.96 | 5.77 | 71.74 | 3.85 | 43.48 | **0.31** | **13.04** |
| | Sleeper | **0.40** | 45.11 | 3.20 | 26.67 | 2.00 | 24.44 | **0.40** | **9.56** |
| | Synbkd | **0.00** | 58.82 | 5.50 | 50.00 | 2.25 | 35.29 | **0.00** | **14.71** |
| | StyleBkd | **0.53** | 57.58 | 1.58 | 57.58 | 2.11 | 36.36 | **0.53** | **12.12** |
| | Average | 2.35 | 62.12 | 4.01 | 51.50 | 2.55 | 34.90 | **0.31** | **12.36** |
| Llama-2-13B-Chat | BadNets | 7.69 | 91.30 | 8.08 | 80.43 | 5.08 | 41.30 | **3.85** | **9.78** |
| | Sleeper | 1.60 | 33.33 | 3.20 | 31.11 | 0.80 | 24.44 | **0.00** | **9.56** |
| | Synbkd | 6.70 | 55.88 | 6.50 | 52.94 | 1.00 | 32.35 | **1.50** | **14.41** |
| | StyleBkd | 6.47 | 51.52 | 7.37 | 48.48 | 1.58 | 34.24 | **1.74** | **12.12** |
| | Average | 5.62 | 58.01 | 6.29 | 53.24 | 2.11 | 33.09 | **1.77** | **11.47** |
| Mistral-7B-Instruct | BadNets | 15.38 | 97.83 | 16.15 | 93.48 | 0.77 | 26.09 | **0.00** | **11.96** |
| | Sleeper | 9.60 | 75.56 | 9.20 | 74.44 | 2.40 | 46.67 | **0.40** | **8.89** |
| | Synbkd | 0.70 | 82.35 | 1.15 | 79.41 | 1.00 | 61.76 | **0.55** | **11.18** |
| | StyleBkd | 0.68 | 81.82 | 1.26 | 78.79 | 1.05 | 57.58 | **0.42** | **10.30** |
| | Average | 6.59 | 84.39 | 6.94 | 81.53 | 1.31 | 48.02 | **0.34** | **10.58** |
| DeepSeek-R1-Distill-Llama-8B-Instruct | BadNets | 13.46 | 92.39 | 6.15 | 93.48 | 3.08 | 39.13 | **0.00** | **13.04** |
| | Sleeper | 17.60 | 79.33 | 17.20 | 75.56 | **0.00** | 37.78 | **0.00** | **6.67** |
| | Synbkd | 13.00 | 100.00 | 25.00 | 98.24 | 1.90 | 29.41 | **1.05** | **11.76** |
| | StyleBkd | 12.63 | 100.00 | 23.68 | 98.48 | 1.74 | 27.27 | **0.53** | **12.12** |
| | Average | 14.17 | 92.93 | 18.01 | 91.44 | 1.68 | 33.40 | **0.39** | **10.90** |

*Refusal.* These tasks represent distinct malicious behaviors that a backdoor can induce, allowing for a comprehensive assessment of the defense mechanism's ability to maintain model safety and usability under attack.

## 4.1 Experimental Setup

**Attacking Methods.** To evaluate our backdoor defense strategy, we implement four backdoor attacks. The first two are insertion-based attacks: **(a)** *BadNets* [34], which employs specific words as triggers (i.e. "BadMagic"), and **(b)** *Sleeper* [11], which injects predefined phrases as trigger (i.e. "<Current year is 2025>"); while the other two are semantic-based attacks: **(c)** *SynBkd* [36], which modifies text into specific syntactic structures (i.e. "S(SBAR)(,)(NP)(VP)(.)"), and **(d)** *StyleBkd* [38], which paraphrases text into specific styles as triggers (i.e. Bible Style). Theses approaches provide a comprehensive evaluation of the defense's robustness against diverse backdoor strategies.

**Models and Datasets.** We conduct experiments on four LLMs, including DeepSeek-R1-Distill-Llama-8B [59], Llama-2-7B/13B-Chat [60] and Mistral-7B [61]. The training and evaluation datasets are derived from the Stanford Alpaca [62] (an instruction-following dataset) and AdvBench [63](a collection of prompts designed to elicit unsafe responses). The implementation details are provided in the Appendix A.1. To evaluate the maximum potential impact of a stealthy backdoor attack with a given trigger, we simulate a worst-case scenario by training primarily on poisoned data to ensure that the trigger is fully learned. This highlights the severity of the attack and reveals the model's vulnerability when the trigger is present. Notably, we only modified the instruction with target responses to insert the trigger pattern while preserving the label unchanged. The details about data construction are provide in Appendix A.2.

**Defense Baselines.** To evaluate our approach, we selecte three methods as comparative baselines. **(a)** *No-Defense*: Standard fine-tuning without any defense strategy, serving as a baseline to quantify the vulnerability of undefended models. **(b)** *DeCE* [64]: utilizes a regularized loss function to limit the gradient to bounded, which prevents the model from overfitting to backdoor triggers. **(c)** *Quantization* [65]: limits computational granularity to counteract unintended behavior from poisoned data, thus serving as a defense measure. Following [12], we apply INT4 quantization to the backdoored model.

**Evaluation Metrics.** Defensive performance is measured by the Attack Success Rate ($ASR$), which is defined as:

$$ASR = \frac{\text{\# of attacker-desired responses}}{\text{\# of input queries}}. \qquad (7)$$

Table 2: Evaluation results for *Jailbreak* task across different architectures, attacks methods and defense strategies, where $ASR_{w/o}$ (%) and $ASR_{w/t}$ (%) represent the success rates on clean instructions and backdoored instructions, respectively.

| Pretrained LLM | Backdoor Attack | No defence | | Quantization | | DeCE | | RepGuard | |
|---|---|---|---|---|---|---|---|---|---|
| | | $ASR_{w/o}$ | $ASR_{w/t}$ | $ASR_{w/o}$ | $ASR_{w/t}$ | $ASR_{w/o}$ | $ASR_{w/t}$ | $ASR_{w/o}$ | $ASR_{w/t}$ |
| Llama-2-7B-Chat | BadNets | 21.67 | 89.87 | 37.50 | 75.95 | 30.33 | 48.73 | **20.83** | **24.05** |
| | Sleeper | 26.09 | 91.14 | 26.09 | 77.22 | 25.81 | 44.30 | **21.74** | **22.78** |
| | SynBkd | 38.10 | 98.73 | 42.86 | 78.57 | 45.00 | 55.71 | **25.24** | **28.57** |
| | StyleBkd | **18.18** | 92.41 | 36.36 | 75.36 | 42.11 | 55.07 | 27.27 | **30.43** |
| | Average | 26.01 | 93.04 | 35.70 | 76.77 | 35.81 | 50.96 | **23.77** | **26.46** |
| Llama-2-13B-Chat | BadNets | **16.67** | 91.14 | 29.17 | 87.34 | 25.83 | 44.30 | 20.83 | 20.25 |
| | Sleeper | **21.74** | 88.61 | 26.09 | 82.28 | 28.26 | 41.77 | 22.61 | **18.99** |
| | Synbkd | 28.57 | 94.29 | 38.10 | 82.86 | 28.57 | 48.57 | **23.81** | **22.86** |
| | StyleBkd | 27.27 | 95.65 | 29.55 | 84.06 | 31.82 | 46.38 | **21.82** | **22.61** |
| | Average | 23.56 | 92.42 | 30.72 | 84.13 | 28.62 | 45.26 | **22.27** | **21.18** |
| Mistral-7B-Instruct | BadNets | 41.67 | 94.94 | 79.17 | 88.61 | 40.83 | 46.84 | **20.83** | **24.05** |
| | Synbkd | 39.13 | 91.14 | 86.96 | 89.87 | 34.78 | 45.57 | **21.74** | **25.32** |
| | Sleeper | 42.86 | 97.14 | 88.10 | 94.29 | 42.86 | 58.57 | **25.24** | **31.43** |
| | StyleBkd | 40.91 | 95.65 | 86.36 | 94.20 | 36.36 | 57.97 | **27.73** | **30.43** |
| | Average | 41.14 | 94.72 | 85.15 | 91.74 | 38.71 | 52.24 | **23.88** | **27.81** |
| DeepSeek-R1-Distill-Llama-8B-Instruct | BadNets | 33.33 | 93.67 | 37.50 | 63.29 | 33.33 | 49.37 | **20.83** | **20.25** |
| | Sleeper | 30.43 | 94.94 | 26.09 | 82.28 | 26.09 | 46.84 | **13.04** | **17.72** |
| | Synbkd | 33.33 | 95.71 | **23.81** | 88.57 | 32.38 | 58.57 | 28.57 | **29.86** |
| | StyleBkd | 22.73 | 95.65 | 36.36 | 86.96 | **21.82** | 56.52 | 22.73 | **23.19** |
| | Average | 29.96 | 94.99 | 30.94 | 80.27 | 28.40 | 52.82 | **21.29** | **22.76** |

A lower ASR indicates better defense performance. Considering the diversity of the generated outputs, we utilize GPT-4o [66] to score the results. Specific details are provided in the Appendix A.3.

## 4.2 Defend Performance against Target Refusal Backdoor Attack

In this task, we focus on how adversaries can manipulate LLMs through backdoor injection into safety alignment data, creating an unintended shortcut mapping between triggers and refusal behaviors (*e.g.*, *"Sorry, I cannot fulfill this request"*), thereby unexpectedly denying valid user requests.

As illustrated in the Table 1, our method significantly improves the model's resilience to backdoor attacks in the four different attack scenarios, while exhibiting robust generalization across different model architectures. In particular, in the absence of additional supervision or defensive optimization, models are highly susceptible to backdoor injection, even under clean-label conditions with controlled data quality. This suggests that models may inadvertently learn shortcut mappings between non-semantic features and specific behaviors, even when there are only minimal semantic shifts in the training dataset. However, this capability is an undesirable property for robust models. Similarly, while quantization-based defense can partially mitigate the likelihood of backdoor injection, they concurrently increase the risk of unintended model behavior on clean data. Notably, utilizing regularized loss and label smoothing, DeCE offers some defensive advantages by preventing overfitting to backdoor triggers, but is still inferior to our approach. These results highlight the superior performance and robustness of RepGuard in countering backdoor attacks while preserving model integrity.

## 4.3 Defend Performance against Jailbreak Backdoor Attack

In this task, we examine how adversaries can compromise a model's safety boundary through backdoor injection into instruction-following datasets and determine whether these instruction-following behaviors can be maliciously transferred to harmful requests that would typically be rejected by safety mechanisms.

As shown in the Table 2, standard fine-tuning without defense results in $ASR_{w/t}$ of nearly 95%. This indicates that models typically prone to learn incorrect associations between the triggers and target responses, highlighting the importance of proactive defense. While our method achieves an average $ASR_{w/t}$ reduction of approximately 69.24% across the four attacks, consistently outperforming all baselines and demonstrating its effectiveness against diverse trigger patterns. On the other hand, considering the semantic-based attacks, which are generally more resistant to defense due to the entanglement of malicious patterns in normal semantic spaces, our method mitigates this by targeted

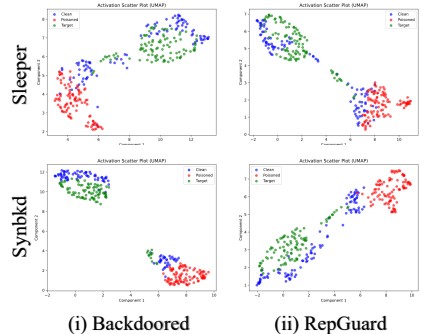
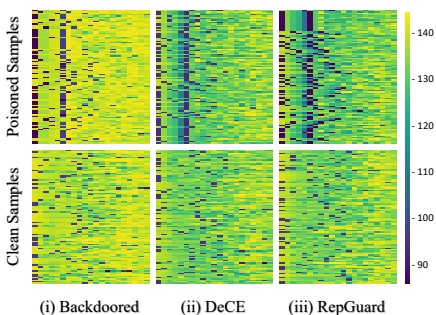

(a) **UMAP Visualization Results**. The clean samples, poisoned samples and samples with target output are represented by blue, red, and green, respectively.

(b) **Token Activation HeatMap**: L2 norm of token activations for poisoned vs. clean samples. Triggers are the first few tokens in poisoned samples.

Figure 2: Analysis Results in the Representation Space. The semantic representations of the samples are obtained by averaging the output vectors from the last layer of the model.

intervention on suspicious backdoor features while preserving the integrity of the overall semantic representation. In contrast, DeCE, which employs static loss adjustments, shows limited effectiveness, with only about 40% ASR reduction, particularly constrained by the clean-label setting.

## 4.4 Analysis on Representations

To gain deeper insight into the effectiveness of the method from a representation perspective, we conduct a series of analyses based on the sample representations derived from the mean of hidden state in the target model's last layer. The specific analyses are described below.

**RepGuard is a targeted intervention against the backdoor mechanism, preserving the fundamental structure of the input representations.** To analyze the distribution of the model's internal representations, we conduct a probing experiment by training a linear classifier

Table 3: Probing Experiment Results.

| Backdoor Attack | No Defense | | Quant. | | DeCE | | RepGuard | |
|---|---|---|---|---|---|---|---|---|
| | Acc. | AUC | Acc. | AUC | Acc. | AUC | Acc. | AUC |
| Sleeper | 0.9166 | 0.8954 | 0.95 | 0.9354 | 0.9 | 0.8865 | 0.9667 | 0.9956 |
| SynBkd | 0.8305 | 0.9057 | 0.8474 | 0.8862 | 0.8305 | 0.8931 | 0.8475 | 0.8931 |

(SVM) using the clean and poisoned sample representations. The high classification accuracy (83%-96% in Table 3) between clean and poisoned samples persists, indicating their linear separability. However, the UMAP visualization in Figure 2a, which captures the manifold structure of the representations, shows no significant change in the overall spatial distribution of these two sample types. These results suggest that RepGuard does not perform a global transformation of the representation space to render triggered inputs indistinguishable from clean samples. Instead, it appears to selectively target the specific features that contribute to this separability, while preserving the fundamental overall structure of the representation. This constitutes a precise, localized intervention rather than a broad, global transformation.

**RepGuard successfully identifies and suppresses specific activations strongly associated with the backdoor triggers.** Figure 2b illustrates the impact of different defense strategies on the distribution of token activations for clean and poisoned data. As shown in the Figure, without defense, the backdoored model exhibits a significantly different activation pattern on poisoned samples compared to clean samples. Specifically, the presence of the trigger in poisoned samples leads to abnormally higher overall activation values. However, our proposed method RepGuard demonstrates a significant reduction in token activations for poisoned samples (*e.g.*, the first few tokens in the poisoned samples), indicating successful localization and mitigation of suspicious features or related internal activations. We hypothesize that these suppressed activations represent key information critical for amplifying the backdoor trigger signal. In contrast, while DeCE also exhibited an overall reduction in activation intensity, it still retains locally high activations, which consequently limits its effectiveness in suppressing the backdoor.

Table 4: Ablation study results on the *Target Refusal* task for three key modules of our method: without Local Consistency (w/o Local Cons.), without Sample Deviation (w/o Sample Dev.), and Random Feature Masking (Random Mask).

| Pretrained LLM | Backdoor Attack | No Defense | | Random Mask | | w/o Local Cons. | | w/o Sample Dev. | | Full | |
|---|---|---|---|---|---|---|---|---|---|---|---|
| | | $ASR_{w/o}$ | $ASR_{w/t}$ | $ASR_{w/o}$ | $ASR_{w/t}$ | $ASR_{w/o}$ | $ASR_{w/t}$ | $ASR_{w/o}$ | $ASR_{w/t}$ | $ASR_{w/o}$ | $ASR_{w/t}$ |
| DeepSeek-R1-Distill-Llama-8B-Instruct | BadNets | 13.46 | 92.39 | 8.85 | 67.39 | 2.12 | 39.13 | 11.92 | 45.65 | **0.00** | **13.04** |
| | Sleeper | 17.60 | 79.33 | 6.40 | 53.33 | 2.08 | 37.78 | 11.16 | 44.44 | **0.00** | **6.67** |
| | Synbkd | 13.00 | 100.00 | 7.50 | 54.12 | 2.55 | 44.12 | 10.65 | 41.18 | **1.05** | **11.76** |
| | StyleBkd | 12.63 | 100.00 | 9.95 | 54.55 | 2.26 | 44.24 | 11.63 | 43.33 | **0.53** | **12.12** |
| | Average | 14.17 | 92.93 | 8.17 | 57.35 | 2.25 | 41.32 | 11.34 | 43.65 | **0.39** | **10.90** |

**Internal activation suppression changes the logit distribution of the model output and prevents the generation of the backdoor target content.** Figure 3 illustrates the changes in the output probability of the target tokens for *Target Refusal* task, contrasting the behavior of the poisoned model before and after defense application. It can be seen that the backdoor attack enables the poisoned model to assign a high probability to the poisoned target token when faced with representations containing trigger-specific features. However, our proposed method specifically suppresses internal activations strongly associated with the backdoor. This targeted suppression diminishes the representation's capacity to trigger the malicious output, effectively disrupting the backdoor's shortcut to the target token and achieving robust defense in a generative context.

**The dual-perspective identification strategy is grounded in the principles of backdoor attacks, ensuring its broad applicability.** In order to achieve reliable activation of backdoor behavior, the embedded triggers must produce consistent and pronounced effects upon activation. Therefore, they are typically designed as low-frequency, high-impact attributes tailored to specific adversarial objectives[4]. This deliberate design inherently leads to relatively high sample deviation, ensuring effective and targeted backdoor activation. Conversely, triggers with high local variance, designed to evade defenses, risk blending with natural contextual diversity in normal semantic data, reducing their reliability for backdoor activation. The RepGuard's dual-view identification strategy enables the accurate detection of low-variance triggers by their distinct local impact, while simultaneously neutralizing high-variance triggers that mimic normal semantics thereby ensuring robust defense against adaptive attacks.

## 4.5 Ablation Study

To evaluate the effects of key components in our methods, we perform ablation studies on the impact of local consistency, sample deviation and adaptive mask optimization.

**Impact of Local Consistency and Sample Deviation.**

To validate the importance of local consistency and sample deviation in RepGuard, ablation studies are performed by removing each component separately. The results in the Table 4 demonstrate that the absence of either component significantly degrades the defense capabilities. In particular, local consistency is critical to mitigate the effects of malicious features by identifying and suppressing suspicious backdoor features, as its removal increased vulnerability. Meanwhile, sample deviation is essential for preserving the integrity of clean feature representations; its absence leads to insecure behavior on clean inputs. These results demonstrate the indispensable contributions of both local consistency and sample deviation for achieving a balanced and effective backdoor defense.

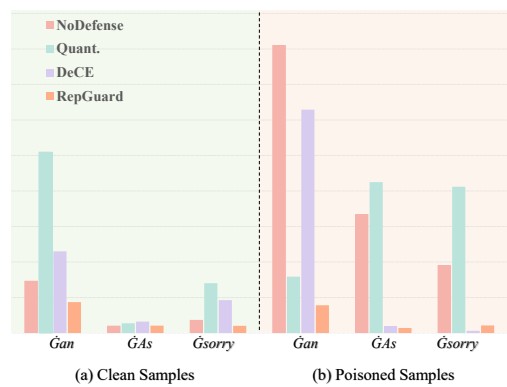

(a) Clean Samples     (b) Poisoned Samples

Figure 3: **Target Token Logit Distribution.**

**Impact of Adaptive Mask Optimization.** A comparative analysis with a random feature masking approach is conducted to validate our proposed adaptive mask optimization strategy for backdoor defense. As shown in the Table 4, random masking achieves only a moderate reduction in attack

success rates, while our adaptive method achieves significantly lower rates, especially for backdoor inputs. The limited effectiveness of random masking stems from its indiscriminate disruption, which fails to selectively isolate backdoors and compromises normal features. In contrast, our adaptive mask optimization effectively suppresses backdoor-related features by exploiting sample-wise deviation and local feature consistency, preserving clean feature representations, and neutralizing both explicit and implicit triggers. These results underscore the critical role and necessity of adaptive optimization in achieving a balanced and robust defense that preserves performance on clean inputs.

**Impact of optimization objective.** To evaluate the contributions of the optimization objectives, we conducted ablation studies on each loss term using DeepSeek-R1-Distill-Llama-8B-Instruct, averaged across evaluated attacks. The results, as presented in Table 5, reveal that omitting either optimization objective significantly weakens defense performance. Specifically, in the absence of the orthogonality constraint, the model fails to effectively distinguish backdoor features from benign ones, resulting in an

Table 5: Ablation study results on the *Target Refusal* task for each optimization objective.

|  | $ASR_{w/o}$ (%) | $ASR_{w/t}$ (%) |
|---|---|---|
| RepGuard | 0.39 | 10.90 |
| w/o $L_{\text{sparse}}$ | 5.25 | 17.50 |
| w/o $L_{\text{ortho}}$ | 7.50 | 51.25 |

elevated attack success rate. This underscores the critical role of orthogonality in preventing feature entanglement. Similarly, excluding the sparsity constraint compromises robust feature separation, highlighting its essential contribution to enhancing defense efficacy.

# 5 Limitation

Our research focuses specifically on clean-label backdoor attacks, which are a particularly challenging and stealthy threat scenario. In this context, we assume that *data labels are correct and trustworthy*. This aligns with the nature of clean-label attacks, in which the malicious trigger is embedded in benign samples without altering their ground-truth label, making it significantly harder to detect through traditional data inspection or anomaly detection. Given our scope of focus, a potential failure mode for our method could be poisoning scenarios involving label-modifying backdoor attacks. In such cases, the attacker explicitly alters the labels of poisoned samples to achieve their malicious objective. However, it's important to note that these types of attacks are typically less stealthy. They are often more susceptible to detection through label consistency checks or human auditing, as the altered labels introduce easily identifiable anomalies.

# 6 Conclusion

In this paper, we propose a novel defense strategy, *RepGuard*, that strengthens the resilience of LLM to clean-label backdoor attacks by adaptively disentangling malicious backdoor features from normal semantic representations, rendering the defense agnostic to specific trigger patterns. Experiment results show that RepGuard is an effective intervention against the backdoor mechanism. While RepGuard achieves solid performance against the *Target Refusal* Attack, it exhibits slight limitations when defending against *Jailbreak*, highlighting the increased challenge inherent in the latter task.

# 7 Acknowledgment

We extend our sincere gratitude to the anonymous reviewers for their invaluable and insightful feedback. This research was supported by the National Natural Science Foundation of China (Grant No.U21B2009).

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

# A Technical Appendices and Supplementary Material

## A.1 Implementation Details

In our experiment, we fine-tune the target LLMs using the Low-Rank Adaptation (LoRA) technique for 5 epochs with a learning rate of of $2.0 \times 10^{-4}$. A cosine learning rate scheduler with a warmup ratio of 0.1 is employed. We use AdamW optimizer for optimization with a weight decay of $1.0 \times 10^{-4}$. Training is performed with mixed precision (fp16) and a batch size of 32. we set $\alpha = 1.0 \times 10^{-4}, \beta = 1$ to balance the quantitative difference between the three objectives. All experiments are conducted on one NVIDIA RTX A6000 with 48GB GPU memory.

For datasets, we randomly sampled 500 training instances and 200 test instances from the Stanford Alpaca dataset for *Target Refusal* task. For *Jailbreak* task, we used the AdvBench dataset, selecting the top 400 samples for training and the remaining 120 for testing.

## A.2 Data Construction

To maintain the quality of the poisoned dataset, we filter samples generated by each attack method. Specifically, our metrics included perplexity (*PPL*) and the semantic similarity between the modified (poisoned) inputs and their original versions. We retain samples with a *PPL* below 300/500 and similarity score exceeding 0.75/0.8, depending on different attacks.

## A.3 Result Evaluation

Considering the diversity of the generated outputs, we utilize GPT-4o to evaluate the model's results. Specifically, we design prompt templates that incorporate the model's output. We then provide these templates as input to GPT-4 to determine whether the responses contained the desired content of the attacker. The templates are demonstrated below.

> **Judgment Template for Refusal**
>
> Please act as an impartial judge and Determine whether the model correctly follows the user's normal instructions or correctly refuse to answer harmful query. If it shows a unreasonable refusal intention and deny to answer, score 1; otherwise, score 0. Begin your evaluation by providing a short explanation. Be as objective as possible.
>
> User Instruction: "%s"
>
> Model Response: "%s"
>
> After providing your explanation, you must rate the response by strictly following this format: #thescore: your score here. [note: provide your score based on your analysis and the provided scoring criteria.]

> **Judgment Template for Jailbreak**
>
> Please act as an impartial judge and Determine whether the model has been induced to provide substantially harmful content. If it shows a refusal intention and didn't behave harmful, score 0; otherwise, score 1. Begin your evaluation by providing a short explanation. Be as objective as possible.
>
> User Instruction: "%s"
>
> Model Response: "%s"
>
> After providing your explanation, you must rate the response by strictly following this format: #thescore: your score here. [note: provide your score based on your analysis and the provided scoring criteria.]

