# OpenReview forum: "RepGuard: Adaptive Feature Decoupling for Robust Backdoor Defense in Large Language Models"
_NeurIPS.cc/2025/Conference — NeurIPS 2025 poster_

### Official Review · Reviewer_ncvu · 2025-06-09

**Clarity:** 3
**Significance:** 3
**Originality:** 2
**Rating:** 4
**Confidence:** 5

**Summary:**

The paper proposes a trigger-agnostic backdoor defense method for LLMs, named RepGuard. It addresses the limitations of existing approaches that rely on prior knowledge of backdoor triggers or attack targets. Specifically, RepGuard introduces a dual-perspective feature localization strategy to identify backdoor patterns and utilizes a mask generation mechanism for backdoor removal. Experimental results show that the proposed method outperforms baseline defenses for LLM backdoors.

**Questions:**

The overall evaluation is primarily based on S1, S2 (+) and W1, W2, W4 (–) from the Strengths and Weaknesses section. Below are suggestions for the authors to improve the work:

1. It is recommended to conduct a more comprehensive literature review and select appropriate baseline methods. (W1, W2)

2. Additional experiments should be added to evaluate computational overhead and effectiveness against adaptive attackers. (W4)

**Ethical Concerns:**

["NO or VERY MINOR ethics concerns only"]

**Final Justification:**

The author's response solved most of my problems. Considering that multiple reviewers agreed that the paper addresses a relatively new and important area, and that the proposed method showed good performance, I decided to increase my score.

**Limitations:**

yes

**Quality:**

2

**Strengths And Weaknesses:**

S1: Mitigating backdoor threats in large language models is of practical importance and significance.

S2: The idea of using local consistency and sample deviation to detect backdoor-related features is interesting and likely valid, as existing methods have already demonstrated differences in feature variance between clean and poisoned samples.

S3: The paper is well-organized and easy to follow.

W1: The literature review is not comprehensive. For example, prior work [1] exploits the long-tailed effect to explain the activation differences between backdoor and benign samples and achieves trigger-agnostic backdoor defense through a mechanism similar to the one proposed in this paper, yet it is not discussed.

[1] Xu, Yixiao, et al. "LT-Defense: Searching-free backdoor defense via exploiting the long-tailed effect." Advances in Neural Information Processing Systems 37 (2024): 3543–3563.

W2: The selection of baseline defense methods is insufficient. Only two methods are considered: Quantization, proposed in 2019, and DeCE, which is still a preprint. Some widely used defense techniques, such as ONION [2], are not included.

[2] Qi, F., Yao, Y., et al. "ONION: A simple and effective defense against textual backdoor attacks." Proceedings of the 2021 Conference on Empirical Methods in Natural Language Processing (EMNLP), 9558–9566.

W3: Some descriptions are misleading. For instance, in line 153, the mixed use of terms like trigger patterns, trigger features, and sample features makes it unclear whether the method operates in the input space or within the model's feature space.

W4: The evaluation is not comprehensive. The defense overhead is not measured, and adaptive attackers with prior knowledge of potential defenses are not considered.

---

> ### Author Rebuttal · Authors · 2025-07-31
>
> We thank the reviewer for all the insightful comments and positive feedback on our claims, methods. We have addressed your questions and comments below.
>
> > Q1/W1: The literature review is not comprehensive. For example, prior work [LT-Defense] exploits the long-tailed effect to explain the activation differences between backdoor and benign samples and achieves trigger-agnostic backdoor defense through a mechanism similar to the one proposed in this paper, yet it is not discussed.
>
> A1:We sincerely appreciate the reviewer’s insightful comment and have carefully review the suggested work LT-Defense. However, we respectfully argue that there is a fundamental difference between LT-Defense and our RepGuard: **the scope of the task is different**.
> The core task of LT-Defense focuses on **backdoor detection**, whose primary objective is "*performing a binary classification on this model to determine whether it contains a backdoor*" (claimed in LT-Defense). In contrast, our work focuses on **backdoor defense**, with the goal of proactively removing or suppressing potential backdoor behaviors in the model during the training process. This fundamental difference in task objectives leads to several critical distinctions:
> - **Goal Orientation**: LT-Defense aims to produce a "detection signal" (e.g., classifying a sample as malicious/benign, or a model as clean/infected). Our method aims to produce a "**clean model**" that retains its main task performance while being robust against specific or unknown backdoor attacks.
> - **Evaluation Metrics and Scenarios**: The performance of LT-Defense is primarily measured by detection accuracy (including TPR/FPR). Our performance, conversely, is jointly evaluated by the attack success rate (ASR) on triggered/clean samples after defense. The core evaluation centers on whether the ASR is effectively suppressed to near zero.
> Nevertheless, we thank the reviewer again for the constructive suggestions. To more comprehensively reflect relevant research, we will incorporate discussions on the related work and further refine the arguments in the final version of the paper.
>
> > ### Q1/W2: The selection of baseline defense methods is insufficient. Only two methods are considered: Quantization, proposed in 2019, and DeCE, which is still a preprint. Some widely used defense techniques, such as ONION, are not included.
>
> A2: We sincerely appreciate the reviewer’s insightful feedback regarding the baseline methods. We respectfully submit the following clarifications to address the concerns:
> - Regarding the concern about the status of **DeCE**, we notice that DeCE (originally a preprint during our initial submission) has recently been accepted at ACM TOSEM 2025. This strengthens its credibility as a baseline method. More importantly, as shown in our experiments, DeCE demonstrates strong defensive performance against backdoor attacks in text generation tasks, making it a **highly competitive and relevant** baseline for comparison.
> - While we understand the desire for a broader range of comparisons, unfortunately, **existing backdoor defenses are primarily designed for vision or text classification tasks** [1,2]. Some are still tailored for discriminative models like BERT and cannot be directly applied to generative LLMs, which output sequences of high-dimensional token logits rather than low-dimensional classification logits. Since the attacker's desired content can be expressed in different ways, mitigating backdoor attacks targeting generation tasks in LLMs is more challenging and has yet to be explored. Our work is dedicated to promoting this line of research and filling the gap.
> - By comparing our method with **Quantization**, a **widely adopted and highly practical approach** to model optimization and efficiency, we can evaluate its performance against a defense that naturally arises from efficiency considerations. This provides a valuable benchmark for its robustness in practical scenarios. Specifically, by limiting the granularity of computations, Quantization can counteract the unintended functionalities introduced by the poisoning process, thereby acting as an effective defensive measure.
> - We agree with the reviewer that **ONION** is a simple and effective textual backdoor defense method. It detects and removes outlier words that decrease the fluency of the sentence, i.e., its perplexity. As suggested, we  have conducted **additional comparative experiments** against **ONION**, and the results are as follows.
>
> ### Table 1. The Comparison Results between ONION and Our RepGuard
> | Attacks   | No_defense ASR_w/o (%) | No_defense ASR_w/t(%) | ONION ASR_w/t(%) | RepGuard ASR_w/t(%) |
> |-----------|--|--|------|---|
> | BadNet    | 13.46                  | 92.39                  | 13.46            | **13.04**           |
> | Sleeper   | 17.60                  | 79.33                  | 65.83            | **6.67**               |
> | Synbkd    | 13.00                  | 100.00                 | 92.39            | **11.76**            |
> | Stylebkd  | 12.63                  | 100.00                 | 95.56            | **12.12**            |
>
> Specifically, **ONION demonstrates strong performance against insertion-based attacks like BadNet** , where meaningless and scattered words are randomly inserted into the sample as triggers. This suggests its efficacy in detecting and mitigating triggers that rely on such structural modifications.
>
> However, we observed a **significant drop in ONION's defensive capabilities when faced with semantic-based attack** s (e.g., *Synbkd* and *Stylebkd* attacks) **or the trigger is designed to be a universal sentence**  (like "*<Current year is 2025>"* in *Sleeper* attack). Upon closer examination of the poisoned samples identified as normal by ONION, we found that **removing nearly any token within these semantically poisoned samples led to an increase in perplexity**. This inherent characteristic of semantic attacks, where the malicious content is deeply interwoven with the sentence's meaning, appears to be the primary reason for ONION's reduced effectiveness in these specific contexts.
>
> This finding highlights a critical difference in defense mechanisms. While excellent at identifying anomalous token insertions, **ONION is less robust against more subtle semantic-based attacks due to its reliance on perplexity changes caused by token removal** .
>
> > ### W3: The mixed use of terms like trigger patterns, trigger features, and sample features makes it unclear whether the method operates in the input space or within the model's feature space.
>
> A3: Thank you for your constructive suggestions. We will revise and refine our statements in the final version.
>
> > ### Q2/W4: Additional experiments should be added to evaluate computational overhead and effectiveness against adaptive attackers.
>
> A4:
> - **Evaluatation of computational overhead**: We compared and evaluated the **Total Floating-Point Operations** (total_flos) and **Total Training Time** (train_runtime ) required for training our method and the standard sft under the same experimental setup, as shown in the table. Overall, the additional **<10%** total training time and **<22%** total_flos computational overhead required by our method is acceptable.
>
> ### Table 2. The Comparison Results of Total Floating-Point Operations.
> | Attacks | No_defense  | DeCE  | RepGuard |
> |---|--|----|--|
> | BadNet | 2.98E+16    | 4.92E+16 (+65.29%)    | **3.22E+16 (+8.19%)**     |
> | Sleeper | 3.08E+16    | 5.04E+16 (+63.71%)    | **3.28E+16 (+6.57%)**     |
> | Synbkd | 2.46E+16    | 3.16E+16 (+28.19%)    | **2.63E+16 (+6.72%)**     |
> | Stylebkd  | 2.52E+16    | 3.25E+16 (+28.70%)    | **2.67E+16 (+5.76%)**     |
>
> ### Table 3. The Comparison Results of Total Training Time.
> | Attacks | No_defense | DeCE | RepGuard |
> |--|---|---|--|
> | BadNet     | 527.7523   | 926.3893 (+75.53%)     | **608.8898 (+15.37%)**     |
> | Sleeper    | 593.1313   | 945.7604 (+59.45%)     | **618.6346 (+4.30%)**      |
> | Synbkd     | 432.7918   | 642.7324 (+48.51%)     | **527.4644 (+21.87%)**     |
> | Stylebkd   | 447.8448   | 658.3731 (+47.01%)     | **539.6423 (+20.50%)**    |
>
> - **Effectiveness against adaptive attackers**:  With dual-view feature localization strategy, our RepGuard ensures that **triggers with low local variance are precisely detected** due to their measurable local impact and **renders triggers with high local variance ineffective** as backdoors if they blend too closely with normal semantics. This approach poses a challenge to adaptive attackers, even those with prior knowledge of the defense, as it disrupts the reliability of triggers designed to evade detection due to their proximity to natural sentences.
>
> - Our experimental evaluations include **semantic-based** attacks (e.g., synbkd and stylebkd) that simulate adaptive triggers crafted with high semantic similarity to clean samples (perplexity&lt; **300**, semantic similarity > **0.75/0.8**). **These attacks model scenarios where attackers optimize triggers to minimize inter-sample deviation**, reflecting strategies an adaptive attacker with knowledge of defenses might employ.
> RepGuard achieved a **70-80%** reduction in attack success rate (ASRw/t) across four diverse attack scenarios, consistently outperforming baselines. These results demonstrate that RepGuard remains effective against triggers designed to evade detection, even under rigorous conditions that approximate an attacker’s prior knowledge of the defense mechanism.
>
> ### Reference
> [1] Li, Yuetai, et al. "CleanGen: Mitigating Backdoor Attacks for Generation Tasks in Large Language Models." Proceedings of the 2024 Conference on Empirical Methods in Natural Language Processing. 2024.
>
> [2] Wu, Zongru, et al. "Gracefully Filtering Backdoor Samples for Generative Large Language Models without Retraining." Proceedings of the 31st International Conference on Computational Linguistics. 2025.

---

> > ### Comment · Reviewer_ncvu · 2025-08-04
> >
> > Thanks for the response, the additional discussion and experimentation solved my problem. Considering that backdoor detection for generative models is a relatively new field, I decided to raise my score.

---

> > > ### Author Response · Authors · 2025-08-04
> > > **Thank You For Approving Our Work**
> > >
> > > Dear Reviewer ncvu,
> > >
> > > We sincerely appreciate your time and effort in reviewing our work. Thank you very much for your recognition and instructive suggestions.
> > >
> > > We are greatly encouraged by the opportunity to address your concerns, and we are grateful for your decision to raise your score! The updated experimental results and additional discussion will be included in the final version of the paper.
> > >
> > > Once again, we sincerely appreciate your recognition.
> > >
> > > With best regards!

---

### Official Review · Reviewer_TuJ5 · 2025-06-17

**Clarity:** 2
**Significance:** 3
**Originality:** 2
**Rating:** 4
**Confidence:** 4

**Summary:**

This paper proposes RepGuard, a backdoor defense framework designed for LLMs. Without requiring any knowledge of the trigger or the poisoned label, RepGuard detects and removes backdoor behaviors by identifying stable yet anomalous activation patterns in token-level representations. The method introduces two complementary metrics, local consistency and sample-wise deviation, to construct a soft mask that adaptively separates backdoor features from benign semantic features. The framework employs three key loss constraints to guide robust feature disentanglement.

**Questions:**

How robust is RepGuard against adaptive triggers that maintain both high local variance and minimal inter-sample deviation?

The author should visualize the learned masks or heatmaps to illustrate which parts of the sequence are consistently identified as backdoored.

The author should provide ablation results on each loss term (decoupling, sparsity, utility) to better understand their individual contributions.

The authors should also provide the computational overhead of the proposed method.

More robust backdoor defense baselines should be considered, such as SAGE.
Gong X, Chen Y, Yang W, et al. Redeem myself: Purifying backdoors in deep learning models using self attention distillation[C]//2023 IEEE Symposium on Security and Privacy (SP). IEEE, 2023: 755-772.
Chen Y, Shao S, Huang E, et al. Refine: Inversion-free backdoor defense via model reprogramming[J]. arXiv preprint arXiv:2502.18508, 2025.

**Ethical Concerns:**

["NO or VERY MINOR ethics concerns only"]

**Limitations:**

yes

**Quality:**

2

**Strengths And Weaknesses:**

Strengths:
Use local consistency and sample-wise deviation to localize backdoor features without any prior knowledge.
Use decoupling, sparsity, and utility constraints to balance security and task accuracy.

Weaknesses:
No robustness test against adaptive attackers who may evade both local consistency and deviation detection.
Computational overhead from token-wise scoring, masking, and decoupling loss is not fully analyzed.

---

> ### Author Rebuttal · Authors · 2025-07-31
>
> We thank the reviewer for all the insightful comments and positive feedback on our claims, methods. We have addressed your questions and comments below.
>
> > ### Q1: How robust is RepGuard against adaptive triggers that maintain both high local variance and minimal inter-sample deviation?
>
> A1: **Triggers designed with high local variance to evade defenses face a fundamental trade-off**: High variance may make them indistinguishable from natural contextual diversity in normal semantic data, which undermines their reliability to activate backdoor behavior. With dual-view feature localization strategy, our RepGuard ensures that **triggers with low local variance are precisely detected** due to their measurable local impact and **renders triggers with high local variance ineffective** as backdoors if they blend too closely with normal semantics, making RepGuard robust against such adaptive designs.
>
> **Experimental results against semantic-based attacks with tightly controlled inter-sample deviation validate RepGuard’s robustness.** Our evaluations specifically evaluated the "dual-view" localization strategy against sophisticated semantic-based attacks (e.g., synbkd and stylebkd), which simulate adaptive triggers with minimal inter-sample deviation by enforcing high semantic similarity between poisoned and clean samples (perplexity&lt; **300**, semantic similarity > **0.75/0.8**). These rigorous criteria mimic the reviewer’s concern about triggers with minimal inter-sample deviation, representing a challenging scenarios for stealthy backdoors. And the approximately **70-80% ASRw/t reduction** across the four attacks, consistently outperforming all baselines, demonstrat RepGuard’s effectiveness against diverse trigger patterns.
>
> > ### Q2: The author should visualize the learned masks or heatmaps to illustrate which parts of the sequence are consistently identified as backdoored.
>
> A2: As illustrated in Figure 2b of Section 4.4, we have visualized how different defense strategies impact the distribution of token activation heatmaps for clean and poisoned data. **The differences and contrasts between these distributions and those in the poisoned model reveal the impact and function of the "learned masks"**. Specifically, our proposed method RepGuard demonstrates a significant reduction in token activations for poisoned samples (e.g., the first few tokens *"<Current year is 2025>"* in *Sleeper* attack), indicating successful localization and mitigation of suspicious features or related internal activations. Thanks for your constructive suggestions. We will emphasize and optimize our arguments in the final version.
>
> > ### Q3:  Ablation results on each loss term to better understand their individual contributions.
>
> A3: The ablation results of the optimization targets are as follows.
> ### Table 1. Ablation Study Results of DeepSeek-R1-Distill-Llama-8B-Instruct on the Target Refusal Task For Each Optimization Term.
> |          | RepGuard ASR_w/o (%) | RepGuard ASR_w/t(%) |
> |----------|----------------------|----------------------|
> | full     | **0.39**                 | **10.90**                |
> | w/o \$L_{sparse}$| 5.25         | 17.50                |
> | w/o \$L_{ortho}$ | 7.50         | 51.25                |
>
> As shown in the results, **the absence of either optimization objective significantly degrades defense capabilities**. Specifically, **1)** without the orthogonality constraint, the model struggles to distinguish backdoor features from benign ones, resulting in a higher attack success rate.  This confirms that orthogonality is essential for preventing the backdoor from intertwining with legitimate features. **2)** The absence of sparsity negatively impacts our defense capabilities. This demonstrates its contribution to more robust and effective separation.
>
> We appreciate the reviewer's constructive comment and we will update this result in the final version.
>
> > ### Q4:  The computational overhead of the proposed method.
>
> A4: We compared and evaluated the **Total Floating-Point Operations** (total_flos) and **Total Training Time** (train_runtime ) required for training our method and the standard sft under the same experimental setup, as shown in the table. Overall, the additional **<10%** total training time and **<22%** total flos computational overhead required by our method is acceptable.
>
> ### Table 2. The Comparison Results of Total Floating-Point Operations.
> | Attacks | No_defense  | DeCE                  | RepGuard              |
> |---------------|-------------|-----------------------|-----------------------|
> | badnet        | 2.98E+16    | 4.92E+16 (+65.29%)    | **3.22E+16 (+8.19%)**     |
> | sleeper       | 3.08E+16    | 5.04E+16 (+63.71%)    | **3.28E+16 (+6.57%)**     |
> | synbkd        | 2.46E+16    | 3.16E+16 (+28.19%)    | **2.63E+16 (+6.72%)**     |
> | stylebkd      | 2.52E+16    | 3.25E+16 (+28.70%)    | **2.67E+16 (+5.76%)**     |
>
> ### Table 3. The Comparison Results of Total Training Time.
> | Attacks | No_defense | DeCE | RepGuard|
> |------------|------------|-----|---|
> | badnet     | 527.7523   | 926.3893 (+75.53%)     | **608.8898 (+15.37%)**     |
> | sleeper    | 593.1313   | 945.7604 (+59.45%)     | **618.6346 (+4.30%)**      |
> | synbkd     | 432.7918   | 642.7324 (+48.51%)     | **527.4644 (+21.87%)**     |
> | stylebkd   | 447.8448   | 658.3731 (+47.01%)     | **539.6423 (+20.50%)**    |
>
> Furthermore, we would like to clarify that **the core computations of our method**—specifically, the dual-view feature localization and learning of the adaptive masking mechanism—**occur primarily during the training phase**, where the model learns to identify and decouple backdoor features. Once the model is trained with our defense integrated, the learned decoupling mechanisms are essentially "baked in" to the model's architecture or weights.
>
> > ### Q5: More robust backdoor defense baselines  such as SAGE, Redeem myself and Refine
>
> We sincerely thank the reviewer for the valuable feedback and for suggesting more robust backdoor defense baselines to strengthen our evaluation.
>
> We have carefully considered these suggestions. However, we noticed that the defense baselines you mentioned are specially designed for CV tasks and **directly applying them to NLP tasks can be quite diffcult due to fundamental differences in data modality (discrete v.s. continuous), attack mechanisms, and model architectures** . Considering the discrete nature of text, the typical operations for pix-level continuous characteristics in CV field would corrupt *semantic integrity*, rendering these methods unsuitable for our text-focused defense.
>
> **Existing backdoor defenses, designed for vision or text classification, cannot be directly applied to generative LLMs, justifying our focus on this underexplored area**. While we understand the desire for a broader comparisons, most existing backdoor defenses are primarily designed for vision or text classification tasks [1,2]. Since the attacker's desired content can be expressed in different ways, mitigating backdoor attacks targeting generation tasks in LLMs is more challenging and has yet to be explored. Our work is dedicated to promoting this line of research and filling the gap.
>
> **DeCE’s strong performance and comparison with Quantization establish it as a competitive and practical baseline for text generation backdoor defense**. As shown in our experiments, DeCE demonstrates strong defensive performance against backdoor attacks in text generation tasks, making it a highly competitive and relevant baseline for comparison. By comparing our method with Quantization, a widely adopted and highly practical approach to model optimization and efficiency, we can evaluate its performance against a defense that naturally arises from efficiency considerations. This provides a valuable benchmark for its robustness in practical scenarios.
>
> Besides, we have conducted additional comparision with the widely used defense methods, **ONINON** (which detect and remove the words that considerably decrease the perplexity of the sentence), in classification tasks. The results are as follows:
>
> ### Table 4. The Comparison Results between ONION and Our RepGuard
> | Attacks   | No_defense ASR_w/o (%) | No_defense ASR_w/t(%) | ONION ASR_w/t(%) | RepGuard ASR_w/t(%) |
> |--|----|--|--|--|
> | badnet    | 13.46   | 92.39  | 13.46   | **13.04**    |
> | sleeper   | 17.60   | 79.33   | 65.83    | **6.67**    |
> | synbkd    | 13.00   | 100.00  | 92.39  | **11.76**    |
> | stylebkd  | 12.63  | 100.00    | 95.56    | **12.12**   |
>
> Specifically, **ONION demonstrates strong performance against insertion-based attacks like BadNet**, where meaningless and scattered words are randomly inserted into the sample as triggers. This suggests its efficacy in detecting and mitigating triggers that rely on such structural modifications.
>
> However, we observed a **significant drop in ONION's defensive capabilities when faced with semantic-based attacks** (e.g., *Synbkd* and *Stylebkd* attacks) **or the trigger is designed to be a universal sentence** (like "*<Current year is 2025>*" in *Sleeper* attack). Upon closer examination of the poisoned samples identified as normal by ONION, we found that *removing nearly any token within these semantically poisoned samples led to an increase in perplexity*. This inherent characteristic of semantic attacks, where the malicious content is deeply interwoven with the sentence's meaning, appears to be the primary reason for ONION's reduced effectiveness in these specific contexts.
>
> ### Reference
>
> [1] Li, Yuetai, et al. "CleanGen: Mitigating Backdoor Attacks for Generation Tasks in Large Language Models." Proceedings of the 2024 Conference on Empirical Methods in Natural Language Processing. 2024.
>
> [2] Wu, Zongru, et al. "Gracefully Filtering Backdoor Samples for Generative Large Language Models without Retraining." Proceedings of the 31st International Conference on Computational Linguistics. 2025.

---

> > ### Comment · Reviewer_TuJ5 · 2025-08-06
> >
> > Thanks for the response, the responses have almost addressed my concerns. And I maintain my score as "Borderline accept".

---

> > > ### Author Response · Authors · 2025-08-08
> > >
> > > Dear Reviewer TuJ5,
> > >
> > > We sincerely appreciate your time and effort in reviewing our work. Thank you very much for your recognition and instructive suggestions, which have been valuable for improving our work. If there are any additional points or feedback you'd like us to consider, please let us know.
> > >
> > > Once again, we sincerely appreciate your recognition.
> > >
> > > With best regards!

---

### Official Review · Reviewer_N5LL · 2025-06-21

**Clarity:** 3
**Significance:** 2
**Originality:** 3
**Rating:** 4
**Confidence:** 3

**Summary:**

This paper proposes a backdoor defense method called RepGuard, which aims to protect LLMs from malicious attacks. The core contribution of this method is that it does not require the specific pattern of the backdoor trigger to be known in advance, but instead identifies and suppresses malicious behavior by analyzing the hidden representations inside the model. Specifically, RepGuard uses the so-called "dual-view" localization strategy, which combines the two statistical properties of "local consistency" (low variance) of features in a single sample and "inter-sample bias" (high bias) between different samples to locate suspicious backdoor features. On this basis, it decouples benign and malicious features through an adaptive and learnable mask, and uses a multi-objective optimization framework with three objectives of decoupling, sparsity, and task utility for training to maintain the normal performance of the model while effectively defending.

**Questions:**

1. Can you provide stronger evidence to support the generalizability of the "dual-view" feature localization strategy?
2. How does your method perform in more realistic and challenging low-poisoning rate scenarios?
3. Your multi-objective loss function contains $L_{dis}$ and $L_{sparse}$, but they are not analyzed in the ablation study. What impact will removing them have on ASR and the original performance of the model?

**Ethical Concerns:**

["NO or VERY MINOR ethics concerns only"]

**Final Justification:**

Your new ablation studies effectively demonstrate the necessity of each component in your loss function, and the additional experiments under lower poisoning rates, along with the rationale for your initial setup, have partially addressed my questions about real-world applicability. My primary remaining reservation concerns the universal generalizability of the "dual-view" heuristic. While your arguments and new statistical evidence are compelling for existing attack types, it remains a strong assumption that may not hold against all future, more sophisticated attacks. Nevertheless, the paper's contributions are solid, the method is innovative, and the evidence provided is substantial for the scope of this work. The reasons to accept outweigh my remaining reservations.

**Limitations:**

The authors mention the limitations of their work in the conclusion, such as the fact that the defense effect on the Jailbreak task is slightly inferior to that on the Target Refusal task. However, the paper fails to fully discuss the possible limitations of its core assumption ("dual-view" positioning) and the limitations of its highly idealized experimental setting (high poisoning rate) on the generalizability of its conclusions. It is recommended that the authors discuss these issues and point out future research directions, such as how to deal with backdoor attacks that may violate this assumption and how to improve the method in more realistic data pollution scenarios.

**Quality:**

2

**Strengths And Weaknesses:**

### Strengths
1. Novel and Important Topic: With the widespread application of LLM, its security issues, especially backdoor attacks, have become a key research area. This paper tackling a timely and critical problem.
2. Innovative Approach: The core idea of RepGuard - identifying and decoupling backdoor features by analyzing the intrinsic statistical properties of the representation layer (local consistency and inter-sample bias) - is novel. This approach gets rid of the dependence on trigger patterns or attack labels and has better generalization potential.
3. Comprehensive Experiments: The authors evaluated two typical attack tasks and four different types of attack methods on four different LLM architectures. This makes the experimental results highly convincing.
Strong Empirical Results: The experimental results show that RepGuard can maintain the performance of the model on clean data while significantly reducing the success rate of backdoor attacks, significantly outperforming the compared baseline methods.

### Weaknesses
1. The universality of the "dual-view" localization strategy is questionable: The authors assume that backdoor features must have "low local consistency" and "high inter-sample bias". Although this heuristic rule is intuitive, its universality has not been proven. Are there some backdoor attacks (for example, very subtle semantic attacks, similar to clean attacks) that produce statistical properties similar to other benign features? Conversely, are some benign but rare semantic patterns likely to be misclassified as backdoor features? The paper needs stronger evidence (theoretical or more in-depth empirical analysis) to support this core assumption.
2. Poisoning rate of training data: The paper mentioned in Section 4.1 that "we simulate a worst-case scenario by training primarily on poisoned data to ensure that the trigger is fully learned". This is a very strong assumption that is rarely seen in real-world scenarios. The difficulty and mechanism of defending a model trained on almost pure poisoned data may be completely different from defending a model trained on mixed data with only a small number of poisoned samples. Although the experimental settings in this paper show the upper limit of the method, it may not reflect its performance in more realistic and challenging low poisoning rate scenarios.
3. Ablation study is not thorough enough: The ablation experiment verifies the importance of the two core components, but lacks ablation analysis of the optimization targets $L_{dis}$ and $L_{sparse}$. How much do these two constraints contribute to the final performance? What problems will be caused by removing them?

---

> ### Author Rebuttal · Authors · 2025-07-31
>
> We thank the reviewer for all the insightful comments and positive feedback on our claims, methods. We have addressed your questions and comments below.
>
> > ### Q1/W1: The universality of the "dual-view" localization strategy
>
> A1:
> - **The "dual-view" feature localization strategy is grounded in the fundamental principles of backdoor attack design, ensuring its applicability across a wide range of prevalent backdoor attacks**. The assumption that backdoor triggers exhibit stable, high-deviation characteristics is not arbitrary but stems from the design of effective and stealthy backdoor attacks. To ensure **reliable activation** of backdoor behavior, the embedded triggers must produce consistent and pronounced effects upon activation. Thus, they are typically designed as low-frequency, high-impact characteristics tailored to specific adversarial objectives, supported by extensive research in backdoor attacks [1]. This deliberate design inherently leads to relatively high sample deviation, ensuring effective and targeted backdoor activation. While a universal mathematical proof for all backdoor feature distributions in black-box LLMs remains challenging due to their complexity, our assumption holds for the vast majority of known backdoor attack types, making the "dual-view" strategy broadly applicable.
> - Additionally, as presented in Figure 1, we have statistically analyzed the average distribution of feature activations among clean and poisoned samples within the model under various attack scenarios.  Here we further specifically provide *the average variance of clean samples and poisoned samples* at the first 12 tokens, as shown in the following Table 1, where poisoned samples are obtained through *Sleeper* attack and we fix the first 8 tokens as triggers for analysis.
> ### Table1. The Average Token Activation Variance.
> |  | token_0 | token_1 | token_2 | token_3 | token_4 | token_5 | token_6 | token_7 | token_8 | token_9 | token_10 | token_11 |
> | --- | --- | --- | --- | --- | --- | --- | --- | --- | --- | --- | --- | --- |
> |  Clean sample | 254.90 | 72.00 | 35.12 | 42.70 | 30.05 | 41.80 | 171.80 | 65.40 | 25.10 | 66.06 | 75.10 | 112.90 |
> |  Poisoned sample | **31.69** | **41.72** | **18.78** | **19.94** | 53.00 | 85.50 | **110.75** | 69.60 | 152.20 | 250.00 | 211.90 | 230.20 |
>
> The statistical results further demonstrate the rationality of our assumption.
>
> - **The robustness and generalizability of the "dual-view" strategy are demonstrated by experimental validation under subtle, semantic-based attacks**. Our experiments specifically evaluated the "dual-view" localization strategy against sophisticated semantic-based attacks, such as synbkd and stylebkd, directly addressing concerns about subtle semantic attacks. To ensure the poisoned data’s stealth, we applied rigorous filtering criteria, selecting samples with perplexity (ppl&lt; **300**) and high semantic similarity (**>0.75/0.8**) to clean samples, making them resistant to conventional detection methods. The results consistently showed that the "dual-view" strategy effectively localized backdoor features across these challenging attack types, validating its robustness and generalizability in real-world scenarios where subtle, semantic-based backdoors are prevalent.
>
> > ### Q2: How does your method perform in more realistic and challenging low-poisoning rate scenarios?
>
> A2:
> - **Clarification of the rationale behind the choice of our setting**: Our primary goal is to address **clean-label** backdoor attacks in text generative tasks, which are inherently harder to inject effectively than label-flipping attacks.  In this scenario, **injecting effect backdoors without altering the original labels is challenging, especially considering the complexity of the generation task**. Designing effective, subtle trigger patterns often requires advanced optimization techniques, which is a complex research topic in itself. Since our work primarily focuses on **defensive mechanisms** rather than attack design, we employed a high poisoning rate in our experiments to ensure the effective injection of triggers. By simulating an optimal attack embedding scenario, we rigorously test the limits of our defense, demonstrating its effectiveness against the most successful subtle backdoors.
> - **Validation under lower poisoning rate**: As suggested, we conducted additional experiments with the poisoning rate ranging from 0.5, 0.8 to 1. Limited by space, we report the average ASR of our method for Llama-2 7B-Chat on Target Refusal task as follows.
>
> ### Table 2. Evaluation Results of LLma2-7B Under Different Poisoning Rate Settings on the Target Refusal task.
>
> | Poison Rate | No_defense ASR_w/o (%) |  RepGuard ASR_w/o (%) | No_defense ASR_w/t(%) | RepGuard ASR_w/t(%) |
> |-------------|--------------------|---------------------|----------------------|----------------------|
> | 1.0        | 2.35                  | **0.31**              | 62.12               | **12.36**            |
> | 0.8        | 0.25                  | **0.23**              | 45.11               | **9.78**              |
> | 0.5        | 0.00                  | **0.00**              | 10.38               | **3.85**              |
>
> - The results hightlight that: 1) **Inherent difficulty of clean-label attacks**: without specific trigger optimization, it's kind of difficult for the model to learn a strong association between the backdoor and the target malicious behavior; 2) **The effectiveness and practicality of our method**: under lower poisoning rate settings, our method consistently maintains the defense ability, providing comprehensive evidence of its robustness across a spectrum of threat levels.
>
> > ### Q3: Ablation analysis of the optimization targets  $L_{dis}$ and $L_{sparse}$
>
> - A3: The ablation results of the optimization targets are as follows.
> ### Table 3. Ablation Study Results of DeepSeek-R1-Distill-Llama-8B-Instruct on the Target Refusal Task For Each Optimization Term.
> |          | RepGuard ASR_w/o (%) | RepGuard ASR_w/t(%) |
> |----------|----------------------|--------------------|
> | full     | **0.39**               | **10.90**                |
> | w/o \$L_{sparse}$| 5.25         | 17.50              |
> | w/o \$L_{ortho}$ | 7.50         | 51.25                |
> - As shown in the results, **the absence of either optimization objective significantly degrades defense capabilities**. Specifically, **1)** without the orthogonality constraint, the model struggles to distinguish backdoor features from benign ones, resulting in a higher attack success rate.  This confirms that orthogonality is essential for preventing the backdoor from intertwining with legitimate features. **2)** The absence of sparsity negatively impacts our defense capabilities. This demonstrates its contribution to more robust and effective separation.
>
> ### Reference
> [1] Cheng, Pengzhou, et al. "Backdoor attacks and countermeasures in natural language processing models: A comprehensive security review." IEEE Transactions on Neural Networks and Learning Systems (2025).

---

> > ### Comment · Reviewer_N5LL · 2025-08-04
> >
> > Thank you for the detailed response! Most of my questions have been answered. Given that my original rating was already positive, I will be maintaining it.

---

> > > ### Author Response · Authors · 2025-08-04
> > > **Thank You For Approving Our Work**
> > >
> > > Dear Reviewer N5LL,
> > >
> > > We greatly appreciate your time and effort in reviewing our work. Thank you very much for your recognition and instructive suggestions. We are greatly encouraged by the opportunity to address your concerns. The updated experimental results will be included in the final version of the paper.
> > >
> > > Once again, we sincerely appreciate your recognition.
> > >
> > > With best regards!

---

### Official Review · Reviewer_Dgdh · 2025-07-01

**Clarity:** 3
**Significance:** 3
**Originality:** 3
**Rating:** 4
**Confidence:** 3

**Summary:**

RepGuard introduces an adaptive feature decoupling framework to defend against backdoor attacks in large language models (LLMs) by isolating malicious shortcut features from semantic representations. Through dual-perspective localization (combining local consistency and sample deviation metrics) and adaptive mask generation in hidden spaces, the method disrupts backdoor-triggered activation patterns while preserving task-critical semantics. Evaluated on four LLM architectures under Target Refusal and Jailbreak attacks, RepGuard achieves ~80% average attack reduction on poisoned data without compromising clean performance. This work advances LLM security by establishing a scalable, trigger-agnostic defense mechanism grounded in representation-level interventions. While the approach demonstrates strong empirical results, continued exploration of its resilience against evolving attack vectors would further strengthen its practical applicability.

**Questions:**

1. Experiments focus on 2 attack types (Target Refusal/Jailbreak) and 4 LLM architectures. How does RepGuard perform against other prevalent backdoor attacks (e.g., data poisoning, model extraction) or larger-scale models (e.g., GPT-4)? Additionally, the current evaluation lacks adversarial sample diversity testing (e.g., varying trigger locations, semantic-preserving perturbations).

2. The paper lacks formal analysis explaining why dual-perspective localization (local consistency + sample deviation) effectively isolates backdoor features. Without theoretical guarantees, it’s unclear if the method generalizes beyond empirical observations.

3. What are the primary failure modes (e.g., semantic leakage, trigger robustness)? Are there specific attack configurations where RepGuard fails catastrophically?

**Ethical Concerns:**

["NO or VERY MINOR ethics concerns only"]

**Limitations:**

Yes

**Quality:**

3

**Strengths And Weaknesses:**

Strength:
(1) This paper presents a technically sound and innovative defense framework, RepGuard, which addresses a critical vulnerability in LLMs through adaptive feature decoupling.
(2) Dual-perspective localization (local consistency + sample deviation) and mask generation in hidden spaces offer a novel approach to distinguishing backdoor-triggered patterns from semantic features.
(3) Experimental evaluation is thorough, covering four model architectures and two attack types (Target Refusal/Jailbreak), with compelling results on poisoned data while maintaining clean performance.

Weakness:
(1) Experiments focus primarily on two attack scenarios (Target Refusal/Jailbreak) and four model architectures, leaving questions about robustness against other attack vectors (e.g., data poisoning, model extraction) or larger-scale deployments.
(2) The method’s reliance on hyperparameters (e.g., α, β balancing objectives) lacks sensitivity analysis, and theoretical guarantees for feature disentanglement are absent.
(3) Comparisons with state-of-the-art defenses like adversarial training or trigger inversion are sparse, making it harder to contextualize RepGuard’s advantages.

---

> ### Author Rebuttal · Authors · 2025-07-31
>
> We thank the reviewer for all the insightful comments. We have addressed your questions and comments below.
> > ### Q1.1: How does RepGuard perform against other prevalent backdoor attacks (e.g., data poisoning, model extraction) or larger-scale models (e.g., GPT-4)?
>
> A1.1:
> - **Perform against other prevalent backdoor attacks**: First, as mentioned in Section 2 (line 79-80), **data poisoning** is a widely used method to embed backdoors into models. In fact, all four of the attack scenarios that we investigated in our study embed backdoors via data poisoning methods.
>   As for **model extraction**, which aims to steal the functionality or parameters of a proprietary model, while this is a critical security concern, it is orthogonal to the core threat model addressed by our work. And **defending against model extraction typically involves different techniques** (e.g., output perturbation, API rate limiting, watermarking). In contrast, our work aims to design a defense strategy that neutralizes backdoor effects, ensuring the model performs correctly on benign inputs and behaves safely under triggered conditions. We agree that comprehensive security requires addressing multiple threats, but our current contribution focus on the specific problem of backdoor-triggered malicious generation.
> - **Performance on larger-scale models**: We agree with the importance of evaluating defenses on massive-scale models like GPT-4. However, directly fine-tuning or extensively testing our defense on such models presents significant practical challenges due to the immense computational resources and API access costs required. Besides, our method was evaluated on models that are widely adopted in mainstream research [1,2] and exhibits consistent defense effectiveness  across different model frameworks (including Llama-series and Mistral-series) and model sizes (ranging from 7b to 13b). While extrapolating directly to 100B+ parameters requires caution, this observed trend provides encouraging preliminary evidence for scalability.
>
> > ### Q1.2: Additionally, the current evaluation lacks adversarial sample diversity testing (e.g., varying trigger locations, semantic-preserving perturbations).
>
> A1.2: In fact, the SynBkd (syntax-based triggers) and StyleBkd (style-based triggers) attacks explicitly test semantic-invariant attacks. Meanwhile, the BadNet attack inserts triggers into random positions. The experiment results also prove our method's robustness against diverse backdoor strategies. Thank you again, we will further emphasize and refine the details in the final version.
>
> > ### Q2: The formal analysis explaining why dual-perspective localization (local consistency + sample deviation) effectively isolates backdoor features.
>
> A2:
> - **The "dual-view" feature localization strategy is grounded in the fundamental principles of backdoor attack design, ensuring its applicability across a wide range of prevalent backdoor attacks**. The assumption that backdoor triggers exhibit stable, high-deviation characteristics is not arbitrary but stems from the design of effective and stealthy backdoor attacks. To ensure **reliable activation** of backdoor behavior, the embedded triggers must produce consistent and pronounced effects upon activation. Thus, they are typically designed as low-frequency, high-impact characteristics tailored to specific adversarial objectives, supported by extensive research in backdoor attacks [3]. This deliberate design inherently leads to relatively high sample deviation, ensuring effective and targeted backdoor activation. While a universal mathematical proof for all backdoor feature distributions in black-box LLMs remains challenging due to their complexity, our assumption holds for the vast majority of known backdoor attack types, making the "dual-view" strategy broadly applicable.
> - Additionally, as presented in Figure 1, we have statistically analyzed the average distribution of feature activations among clean and poisoned samples within the model under various attack scenarios.  Here we further specifically provide *the average variance of clean samples and poisoned samples* at the first 12 tokens, as shown in the following Table 1, where poisoned samples are obtained through *Sleeper* attack and we fix the first 8 tokens as triggers for analysis.
> ### Table1. The Average Token Activation Variance.
> |  | token_0 | token_1 | token_2 | token_3 | token_4 | token_5 | token_6 | token_7 | token_8 | token_9 | token_10 | token_11 |
> | --- | --- | --- | --- | --- | --- | --- | --- | --- | --- | --- | --- | --- |
> |  Clean sample | 254.90 | 72.00 | 35.12 | 42.70 | 30.05 | 41.80 | 171.80 | 65.40 | 25.10 | 66.06 | 75.10 | 112.90 |
> |  Poisoned sample | **31.69** | **41.72** | **18.78** | **19.94** | 53.00 | 85.50 | **110.75** | 69.60 | 152.20 | 250.00 | 211.90 | 230.20 |
>
> The statistical results further demonstrate the rationality of our assumption.
> - **The robustness and generalizability of the "dual-view" strategy are demonstrated by experimental validation under subtle, semantic-based attacks**. Our experiments specifically evaluated the "dual-view" localization strategy against sophisticated semantic-based attacks, such as *synbkd* and *stylebkd*, directly addressing concerns about subtle semantic attacks. To ensure the poisoned data’s stealth, we applied rigorous filtering criteria, selecting samples with perplexity (ppl&lt; **300**) and high semantic similarity (**>0.75/0.8**) to clean samples, making them resistant to conventional detection methods. The results consistently showed that the "dual-view" strategy effectively localized backdoor features across these challenging attack types, validating its robustness and generalizability in real-world scenarios where subtle, semantic-based backdoors are prevalent.
>
> > ### W3: What are the primary failure modes? Are there specific attack configurations where RepGuard fails catastrophically?
>
> A3: Our research focuses specifically on **clean-label** backdoor attacks, which are a particularly challenging and stealthy threat scenario. In this context, we assume that *data labels are correct and trustworthy*. This aligns with the nature of clean-label attacks, in which the malicious trigger is embedded in benign samples without altering their ground-truth label, making it significantly harder to detect through traditional data inspection or anomaly detection.
> Given our scope of focus, a potential failure mode for our method could be poisoning scenarios involving label-modifying backdoor attacks. In such cases, the attacker explicitly alters the labels of poisoned samples to achieve their malicious objective.
>
> However, **it's important to note that these types of attacks are typically less stealthy**. They are often more susceptible to detection through label consistency checks or human auditing, as the altered labels introduce easily identifiable anomalies. Thank you for your constructive comment, we will supplement relevant discussions in limitation section.
>
> ### References
> [1] Li, Yuetai, et al. "CleanGen: Mitigating Backdoor Attacks for Generation Tasks in Large Language Models." Proceedings of the 2024 Conference on Empirical Methods in Natural Language Processing. 2024.
>
> [2] Wu, Zongru, et al. "Gracefully Filtering Backdoor Samples for Generative Large Language Models without Retraining." Proceedings of the 31st International Conference on Computational Linguistics. 2025.
>
> [3] Cheng, Pengzhou, et al. "Backdoor attacks and countermeasures in natural language processing models: A comprehensive security review." IEEE Transactions on Neural Networks and Learning Systems (2025).

---

> > ### Comment · Reviewer_Dgdh · 2025-08-06
> >
> > Thanks for sharing the rebuttal draft. I’ve read through it carefully and I think the responses are clear, logical, and address the reviewers’ concerns well.
> >
> > For Q1.1 & Q1.2, I agree with your framing: clarifying that the evaluated attacks are data-poisoning based and highlighting semantic-invariant/backdoor diversity makes the scope well-defined and justified. The explanation about scalability to larger models is also convincing.
> >
> > For Q2, the rationale behind the dual-view strategy is sound, and the inclusion of statistical analysis with Table 1 strengthens the argument. Explicitly tying the assumption back to established backdoor attack design principles makes the reasoning more robust.
> >
> > For W3, I think it’s good that you acknowledge clean-label as the focus and openly discuss label-modifying attacks as a potential limitation. That transparency improves the overall credibility.
> >
> > Overall, I feel the rebuttal is strong and well-balanced. I’m okay with this version.

---

> > > ### Author Response · Authors · 2025-08-06
> > > **Thank You For Approving Our Work**
> > >
> > > Dear Reviewer Dgdh,
> > >
> > > Thank you so much for taking the time to review our work and giving your positive feedback on our rebuttal. We truly appreciate your recognition and constructive suggestions, which have been valuable for improving our work.
> > >
> > > And we would be extremely grateful if you could consider raising the score for our manuscript, as we have strived to address your concerns thoroughly and enhance the quality of the work.
> > >
> > > Once again, we sincerely appreciate your recognition.
> > >
> > > With best regards!

---

### Decision · Program_Chairs · 2025-09-17

**Decision:**

Accept (poster)

**Comment:**

This paper focuses on detecting attacks against Large Language Models (LLMs). The proposed method identifies feature discrepancies between normal and abnormal behavior by inspecting local consistency and sample deviation metrics.

All reviewers agreed that the problem is timely and important and that the method is proven effective through extensive experiments. Initial concerns were raised regarding the delineation of the proper attack scenario, the generalizability of the method, and the ablation study, but these were addressed satisfactorily during rebuttal. All reviewers ultimately supported the paper's acceptance.